# Using Chiplet Encapsulation Technology to Achieve Processing-in-Memory Functions

**DOI:** 10.3390/mi13101790

**Published:** 2022-10-20

**Authors:** Wenchao Tian, Bin Li, Zhao Li, Hao Cui, Jing Shi, Yongkun Wang, Jingrong Zhao

**Affiliations:** 1School of Electro-Mechnical Engineering, Xidian University, Xi’an 710000, China; 2Shanghai Sharetek Technology Co., Ltd., Shanghai 200000, China

**Keywords:** Von Neumann structure, Chiplet, processing-in-memory, 2.5D, 3D packaging

## Abstract

With the rapid development of 5G, artificial intelligence (AI), and high-performance computing (HPC), there is a huge increase in the data exchanged between the processor and memory. However, the “storage wall” caused by the von Neumann architecture severely limits the computational performance of the system. To efficiently process such large amounts of data and break up the “storage wall”, it is necessary to develop processing-in-memory (PIM) technology. Chiplet combines processor cores and memory chips with advanced packaging technologies, such as 2.5D, 3 dimensions (3D), and fan-out packaging. This improves the quality and bandwidth of signal transmission and alleviates the “storage wall” problem. This paper reviews the Chiplet packaging technology that has achieved the function of PIM in recent years and analyzes some of its application results. First, the research status and development direction of PIM are presented and summarized. Second, the Chiplet packaging technologies that can realize the function of PIM are introduced, which are divided into 2.5D, 3D packaging, and fan-out packaging according to their physical form. Further, the form and characteristics of their implementation of PIM are summarized. Finally, this paper is concluded, and the future development of Chiplet in the field of PIM is discussed.

## 1. Introduction

In recent years, the rapid development of data centers, AI, 5G, and HPC has increased the demand for computer computing power. However, the traditional von Neumann architecture makes the computing chip separate from the memory separated physically. When accessing memory to process data, it often takes a great deal of time as well as energy, which can lead to the so-called “storage wall” problem [1]. For example, as shown in Figure 1 [1], the CPU rate is 92 TB/s, while the cache rate and bus bandwidth are 1 TB/s and167 GB/s. The data transfer between CPU and off-chip memory consumes two orders of magnitude more energy than floating-point operations [2], which severely limits the overall chip performance. At the same time, with the slowdown of Moore’s Law, to meet the requirement of doubling the computing power every two years, it is necessary to add additional chip size, which increases the manufacturing cost and also reduces the chip yield. Therefore, it is becoming less and less cost-effective to improve chip performance and reduce power consumption by upgrading the IC process [3]. To efficiently process such large amounts of data and break up the “storage wall”, it is necessary to develop PIM technology [4,5]. The core of this architecture is to embed computational power in memory to reduce the high latency and power consumption generated by data movement and to solve the “storage wall” problem [6]. In recent decades, some PIM architectures have been proposed, and their performance has been improved by tens, hundreds, or even thousands of times compared to the traditional von Neumann architecture [2,7,8,9].

PIM architecture in the general sense is to embed the computing power directly in the memory, mainly using the physical relationship between resistance and current and voltage to express the computing process. However, limitations in memory technology have prevented mass production of this PIM architecture. The emergence of Chiplet shows another way to implement PIM functions. Chiplet is a heterogeneous integration technology for small-scale hard IP. It can break a single chip into many small chips, and then each cell can be manufactured by selecting the most suitable process node. These small chips form dense, ultra-short-distance interconnections through advanced packaging technologies, such as 2.5D/3D. The core idea of achieving PIM functionality through Chiplet is to use 2.5D/3D packaging technology to achieve dense, close interconnections between computing Chiplets and memory Chiplets. By shortening the interconnection distance between chips, high-bandwidth packaging links are realized to increase the memory access bandwidth and alleviate the “storage wall” problem.

The purpose of this paper is to summarize the form and characteristics of PIM implementation in the encapsulation field through Chiplet and to provide guidance for development of PIM technology. First, this paper introduces the current development status and the road of PIM technology. Second, Chiplet encapsulation technology that can realize PIM function is introduced and divided into 2.5D, 3D, and fan-out packages according to its physical form. At the same time, the form of the implementation of PIM and the results achieved are introduced and compared. Finally, this paper is concluded, and the future development of Chiplet in the field of PIM is discussed.

## 2. Present Situation of Processing-in-Memory Technology

The basic concept of PIM can be traced back to the 1970s when Kautz et al. first proposed the concept of a memory-computing integration computer in 1969 [10,11]. PIM is considered a promising solution to providing high bandwidth for big data problems and has already achieved speed improvements of several orders of magnitude over many applications: for example, the PIM architecture for graphics processing, including Tesseract and GraphPIM. In addition, there are NNPIM and CMP-PIM for neural network acceleration, etc. [12,13,14,15]. The current research direction of PIM technology is mainly based on volatile memory and non-volatile memory. In addition, combining memory components and processor cores in heterogeneous architectures through Chiplet technology is becoming a mainstream design approach.

Win-San Khwa et al. proposed a 6T SRAM-based PIM architecture and showed that the architecture was able to achieve 57.85 TOPS/W energy efficiency and high throughput of 182.2 GOPS [16]. However, 6T SRAM suffers from reduced read noise margin (RNM) when activating multiple rows, which limits its use in-memory computing and leads to high VDDmin. Qing Dong et al. proposed a 4+2T SRAM that offers searching and logic functions [17]. It uses the N-well as a write WL, eliminating the access transistors and resulting in a 4T-core memory cell. Decoupled read paths enable reliable multi-word activation for in-memory Boolean logic functions. Kim et al. designed 8T SRAM-based Z-PIM architecture for neural that can support zero-skipping operations and fully-variable weight bit-accuracy [18]. The throughput is 66.1% higher than that of von Neumann architecture, and the average power consumption of the Z-PIM chip manufactured in a 65 nm process is 5.294 mW. SRAM-PIM has faster write speeds and lower write energy while maintaining high endurance. In addition, the simple operation mode and mature preparation technology make SRAM one of the directions to design PIM. However, the large cell area of SRAM limits its storage density and hinders its wide application in the PIM field.

Seshadri et al. proposed Ambit, a PIM architecture that operates entirely in DRAM for bulk bitwise [19]. Compared to the most advanced computing systems of the time, it improves performance by 32× and reduces energy consumption by 35×. Shuangchen Li et al. designed DRISA, a DRAM-based PIM architecture, to provide powerful computing power and high bandwidth [20]. Compared with ASICs, it enables 8.8× faster operation and 1.2× higher energy efficiency. Angizi et al. proposed a reconfigurable PIM architecture called ReDRAM, which develops a dual row activation mechanism (DRA) on top of a DRAM subarray [21]. It achieves an average of 54× and 7.1× throughput, respectively, when performing bulk bit-value operations. Although DRAM has high capacity and low cost, the available DRAM IP core process nodes are not advanced, while read latency is high and data need to be refreshed periodically. This, in turn, limits the applicability and programmability of DRAM in the PIM field.

The resistive storage principle of non-volatile memory can provide inherent computational power so that data storage and data processing functions can be integrated with the same physical cell address at the same time. In 2016, Professor Yuan Xie’s team proposed an RRAM-based PIM architecture called PRIME to accelerate neural networks [8]. The team’s experimental results show a performance improvement of about 2360 times and an energy reduction of about 895 times in machine learning benchmarks compared to the most advanced neural processing units at the time. In 2017, Fang Su et al. introduced the first RRAM-based PIM chip, achieving an energy efficiency of 462 GOPs/J at 20 MHz and a 13-fold performance improvement compared to state-of-the-art neural processing units at that time [22]. Shaahin et al. proposed a PIM accelerator (AlignS) for DNA short-read alignment using SOT-MRAM as computational memory and transforming it into a basic processing unit for short-read alignment [23]. It achieves 12× and 1.6× throughput per watt compared to the ASIC accelerator and FM index-based ReRAM platforms of the time, respectively.

Although many PIM architectures based on volatile memory and non-volatile memory have been proposed, it still has limitations that make the architecture of the PIM not able to be commercially produced. For example, the different manufacturing processes of memory and processors do not yet allow for a good balance between processing performance and storage capacity. The non-volatile memory corresponding manufacturers and processes are not yet mature, and there is still a certain distance from real commercialization. Moreover, PIM is sensitive to unbalanced workloads and conflicts as well as other multi-core architectures [24].

The emergence of Chiplet technology provides a new way to realize the function of PIM. Memory and processor cores are integrated into the same package by Chiplet using advanced packaging technology. A typical representative of this type of PIM architecture is the AMD Zen series CPUs [12,25,26]. As shown in Figure 2a, the “Rome” CPU built on Zen 2 contains up to eight CCD Chiplets paired with a single IO-die Chiplet [26]. The CPU contains up to 64 cores and has a total of 256 MB of L3 cache. Nowadays, more and more products have proven that PIM through Chiplet is a viable path, such as Intel’s Stratix 10 FPGA, Agilex series FPGA, AMD’s latest third generation EPYC, Nvidia’s MCM-GPU, France CEA’s 96-core processor, etc. An important requirement for implementing PIM functionality with Chiplet is to optimize the die-to-die interconnect. The interconnect needs to support a variety of product configurations while meeting low power, high bandwidth, and low latency metrics. At this stage, 2.5D, 3D, and fan-out are the packaging methods that have been proven to work for Chiplet interconnects. In 2019, Nurvitadhi et al. developed a small Chiplet-based TensorRAM chip that provides low-latency, high-performance, and high-bandwidth memory and near-memory computational units [27]. They integrated TensorRAM with Intel’s FPGA Stratix 10 using the 2.5D package and compared the performance with Volta GPU. The results show that, for a given computational task of FP32, INT8, the Chiplet-based FPGA and GPU have 57% and 6% of their peak computational power, respectively, and that the latency and energy efficiency of the FPGA are 1/16 and 34× that of the GPU, respectively. In 2017, Vijayaraghavan et al. designed an accelerated processing unit (APU) for tens of billions of computations, which has a modular Chiplet design for its overall architecture [28], as shown in Figure 2b. It integrates GPU and CPU and large-capacity 3D DRAM; with one 3D DRAM on each GPU, the processor uses a total of 8 3D DRAM stacks, which can provide up to 256 GB of capacity and achieve a total bandwidth of 4 TB/s at 1 GHz and 160 w of power consumption. Moreover, the Chiplet-based design of PIM has the following advantages: (1) Chiplet has a smaller area and higher yield, thus reducing production costs; (2) Chiplet can combine heterogeneous structures, saving significant NRE costs and time-to-market; (3) it can also improve system scalability by flexibly combining Chiplet modules.

## 3. Chiplet-Based Processing-in-Memory Architecture

Figure 3a shows a representative 3-dimensional diagram of a Chiplet-based PIM architecture. It integrates multiple small Chiplets via silicon interposers or organic substrates. The PIM architecture includes the memory units, the computing chips, and DRAM, allowing them to be integrated into a large system. Arunkumar et al. divided the GPU into easily fabricated basic GPU Chiplets, with stacked DRAMs distributed around each GPU Chiplet, as shown in Figure 3b [29]. The results show that Chiplet-based GPUs can achieve a 22.8% speed increase with an energy efficiency of 0.5 PJ/bit. Y. S. Shao et al. designed a Simba architecture for deep neural networks (DDN) using Chiplet, as shown in Figure 3c, with 36 Simba packaged together [30], with peak performance of 4 TOPS per Chiplet, up to 128 TOPS, and 6.1 TOPS/W for 36 Chiplets packages. However, for Chiplet-based PIM architectures, the average number of intermediate Chiplets between the source and destination increases with the number of Chiplets, which introduces some latency to the communication. This naturally requires an efficient scaling mechanism to increase the computational power according to the needs of the application. The key consideration for Chiplet-based PIM architectures is the interconnection structure between Chiplets. The interconnection structure accommodates the ability to move high-bandwidth information between clusters of computing cores and memory, as well as the I/O density of transfers from Chiplet to Chiplet. Meanwhile, recent advances in packet-level signaling have led researchers to focus on using Chiplets to implement PIM. Dense integration of memory Chiplets with computing Chiplets can be achieved in three ways: (1) a separate core chip connected to one or more memory chips via silicon inserts (2.5D integration technology); (2) stacking memory chips on top of a core chip (3D integration technology); (3) Chiplets integration using fine-pitch high-density redistribution metal layers (RDL) (fan-out integration technology), allowing formation of almost any size architecture by laying as well as vertical stacking. The attractiveness of Chiplet provides potential unity for advanced PIM architectures. Chiplet-based PIM architectures will be built on reconfigurable systems and used with advanced packaging technologies, such as 2.5D/3D, to significantly increase Chiplet’s potential bandwidth. Chiplet-based PIM architecture is a key research area.

### 3.1. Chiplet on 2.5D Package

To meet the interconnection requirements, fine-pitch routing and bonding techniques should be used. Chip-on-wafer-on-substrate (CoWoS) technology has been developed to provide a good solution for fine pitch and high-density integration [31,32]. The use of adapter plates allows the micro-bump density to be an order of magnitude higher than the package substrate, providing higher interconnect density for the chip and thus high data bandwidth in a smaller area and power consumption. The remaining non-critical signals, such as power supply, are accessed through the C4 bump, as shown in Figure 4a. CoWoS implements PIM in the form shown in Figure 4b, which integrates logical Chiplets with HBM in 2.5D to achieve proximity interconnection [33]. Compared to other packaging solutions, such as multi-chip modules (MCMs), silicon adapter plates provide dense and short metal traces between divided top-level Chiplets and memory stacks. In 2016, NVIDIA introduced the GPU GP100; the GP100 packages the GPU with four HBMs via CoWoS, as shown in Figure 4c, with 16 GB of HBM2 built-in that provides up to 717 GB/s of memory bandwidth [34]. Its computational performance reaches 5.2 TFlops, 10.3 TFlops, and 20.7 TFlops under FP64 double precision, FP32 single precision, and FP16 half-precision floating point, respectively. In 2017, Xilinx introduced a Virtex^®^ UltraScale+™ HBM FPGA, an FPGA that utilizes CoWoS to integrate HBM with the FPGA in the architecture shown in Figure 4d [35]. It offers the highest on-chip memory density, with up to 500 Mb of overall on-chip integrated memory and up to 16 GB of HBM2 integrated into the package, enabling 460 GB/s of memory bandwidth. Breaking up larger chips into Chiplets can reduce defects and thus improve chip yields while also achieving higher mask field utilization. In addition, both the logical Chiplets and HBM in the above architecture can be manufactured on their optimal process nodes. CoWoS technology has been updated, and the amount of HBM storage capacity available has grown, with the fifth generation already offering eight times the total capacity of the second generation [36]. This proves that CoWoS provides an efficient platform for the integration between high-performance logic Chiplets and HBM.

Chiplets connection with low roughness and small wire spacing can also be achieved with embedded multi-die interconnect bridge (EMIB) technology [37,38,39]. Figure 5a depicts a physical construction of the EMIB package, where the silicon bridge is embedded in the package substrate to create a localized, high-density chip interconnect channel. Compared to the silicon-adapter-based 2.5D [33] technology, EMIB avoids the use of large adapter boards and TSVs, further reducing manufacturing costs. At the same time, it enhances signal integrity and reduces structural complexity by creating localized, ultra-high-density interconnects. In this architecture, multiple Chiplets are reorganized by EMIB to optimize the performance and power consumption of the system. For example, EMIB can provide interconnect density up to 300 IO/mm, which can separate the high-speed SerDes IO Chiplet from the core logic Chiplet and reduce the power requirement of high-speed SerDes. As a result, Chiplet can also support domain-specific customization by combining processor core and memory with EMIB technology, which can improve bandwidth and signal quality and alleviate storage wall problems. In Intel’s Agilex series of FPGAs, HBM2e Chiplets are integrated by EMIB next to the FPGA fabric, enabling data transfer rates up to 116 Gbps [40]. Compared to Stratix 10 FPGAs, performance is improved by 45% and power consumption is reduced by 40%, as shown in Figure 5b. It greatly reduces the interconnect distance between the core fabric and the memory compared to a standalone memory, such as DDR.

What is clear is that, instead of stacking the logic Chiplets vertically with the memory, the PIM architecture based on the 2.5D package is placed side-by-side on the adapter board. This allows the bandwidth between the logical Chiplets and the memory to be independent of the interconnect density allowed by the underlying PCB. For example, in 2015, AMD launched a graphics card called the Radeon R9 Fury, with four logic die-controlled 4 HBMs packaged around the GPU in a 2.5D form factor [41]. It can achieve up to 4090 bit memory bit width and can provide up to 512 GB/s memory bandwidth at 1000 MHz, providing the game performance a significant boost. The relatively small area of Chiplet has the advantages of high yield and low cost, in addition to the fact that each Chiplet can be manufactured at its suitable process node, so the 2.5D-based Chiplet architecture greatly reduces the manufacturing cost. At the same time, the 2.5D-based Chiplet architecture performs well in terms of thermal management and mechanical performance, thus improving the overall reliability of the system. Together, these factors can significantly improve the operational efficiency and cost-effectiveness of the system. However, as the system expands, the number of interconnecting interfaces required increases with the number of Chiplets, and interconnect interfaces between Chiplets typically take up more area as well as power overhead than Chiplet interconnects.

### 3.2. Chiplet on 3D Package

TSV-based 3D stacking technology is proven to enable very-high-bandwidth and low-latency memory processor interconnects [42], and the resulting system is well suited for high throughput applications. For example, Kgil et al. proposed a server called PicoServer, which utilizes 3D stacking to interstack a logic chip with multiple DRAM memory chips [43], as shown in Figure 6. The server is capable of running at 500 MHz and delivering 1 Gbps of bandwidth in 3 W of power. Lim et al. developed a 3D multi-core processor 3D-MAPS, a massively parallel processor with 3D stacked memory [44]. The processor is capable of delivering 70.9 GB/s of bandwidth at a maximum operating frequency of 277 MHz with negligible inter-chip data transfer latency and only 4 W of power consumption. However, with the development of HPC applications, the amount of data to be processed is increasing day by day. The emerging 3D packaging technology has the inherent ability to reduce wire length. Further, 3D chip architecture with Chiplet co-designed computing architecture has higher memory bandwidth to alleviate “storage wall” issues. In 2021, AliDAM announced the successful development of a DRAM-based 3D bonded stacked PIM chip. The chip can achieve a throughput efficiency of 184.11 QPS/W and has an on-chip memory density of 64 Mb/mm^2^ and a bandwidth density of 2.4 GB/s/mm^2^ [45]. Therefore, more and more PIM architectures are designed as Chiplet-based 3D structures to improve the performance of computing systems through the co-design of 3D structures and advanced packaging technologies.

Intel’s Lakefield processor combines heterogeneous 3D chip stacking with hybrid computing through Foveros [46]. As shown in Figure 7, the Chiplets are stacked face-to-face on an active silicon adapter board. The performance of Lakefield is 24% more energy-efficient than Core i7-8500Y, while there is up to 91% improvement in power consumption in the former [47,48]. In this architecture, the μBumps used have a pitch of 50 μm and a diameter of 20 μm, which can provide shorter interconnect distances [49]. The ultra-short interconnect length reduces heat generation and parasitic capacitance/impedance, which helps reduce power consumption to achieve high-speed connections with high bandwidth. Lakefield achieves high performance, low power, and small form factor by combining hybrid computing and 3D chip stacking and scaling PC clients to mobile size for low standby power in mobile devices. 

Foveros technology is well compatible with EMIB, and the combination of the two is called Co-EMIB. Co-EMIB allows for higher computational performance and data exchange capabilities, as shown in Figure 8a [50]. The Ponte Vecchio GPU introduced by Intel last year consists of two independent GPU chips as a heterogeneous processor with four HBM2 stacks attached to each GPU module and vertically [51]. The main GPU uses Foveros technology to connect the GPU compute unit to the cache, and EMIB to connect HBM2 and XeLink I/O to the main GPU, as shown in Figure 8b. Micro-bump arrays with 36 μm spacing are used for stacking between the compute and memory modules in this architecture. The denser interconnect distance improves assembly yields and high-power bump density compared to the 50 μm pitch micro-bumps used in Lakefield. Ponte Vecchio is currently in version A0, with measured FP32 throughput performance of over 45TFlops, Memory Fabric cache bandwidth of over 5 TB/s, and interconnect bandwidth of over 2 TB/s. Co-EMIB combines the advantages of 2.5D and 3D to achieve high-density interconnections in both horizontal and vertical directions. This allows Co-EMIB-based systems to achieve higher performance without limiting the flexibility of the system.

Combining the advantages of packages such as EMIB and Si adapter boards, Intel continues to introduce omni-directional interconnect technology (ODI) [52]. It provides greater flexibility for all-around interconnect communication between small- and medium-sized chips in a package and is still in the development stage. The ODI packages offer three types of direct interconnects: (1) chip-to-chip direct interconnects for high bandwidth, high scalability, and short channel distances; (2) top-level chip-to-package substrate direct interconnects for direct power supply and reduced chip I/O parasitic effects; (3) bottom-level chip-to-package substrate direct interconnects. Conventional 3D stacking uses TSVs to achieve high bandwidth communication between the top and bottom chips but is limited by the increased TSV area and heat, as shown in Figure 9a. The passive Si-adapter-based 2.5D technology allows for better thermal management of the processor and memory, but it also reduces communication bandwidth and increases power, as shown in Figure 9b. By optimizing using the ODI as shown in Figure 9c, it is possible to maintain good thermal performance and minimum TSV area for 2.5D while still obtaining the same bandwidth benefits as 3D stacking.

Compared to 2.5D packages, vertically stacked 3D packages enable Chiplet’s bandwidth advantage to be more fully utilized. By stacking multiple Chiplets together using 3D packaging, the size of the chip can be greatly reduced. In addition, the integration of proven Chiplets into new packages can significantly shorten the product development cycle, and Chiplets can be configured in various patterns according to product requirements. This provides higher flexibility and scalability for Chiplet-based architectures compared to traditional SoC computing systems. However, with the stacking of Chiplets, the thermal management of the chips has become a serious challenge in the limited package space. Moreover, the cost and time of chip assembly and testing are higher than that of individual chips.

### 3.3. Chiplet on Fan-Out Packaging

The 2.5D/3D Chiplet architecture based on silicon adapter boards provides enough fine pitch to accommodate high-speed connections with high bandwidth. However, the thickness of the wires on the adapter board is not ideal to support large switching currents. Fan-out packaging technology utilizes high-density redistributed layers (RDL) for integration between Chiplets, enabling flexible and efficient computing systems. As shown in Figure 10a, fan-out chip on substrate (FOCoS) technology with ultra-high I/O density is also capable of memory and computation integration [53]. FOCoS is a fan-out package flip chip mounted on a high-pin-count BGA substrate with a fan-out package RDL. It allows shorter D2D interconnects to be built between multiple chips, which can be used to package application-specific integrated circuits (ASICs) and HBMs. In 2018, Chong et al. introduced a high-density interconnect between ASICs and HBMs using embedded fine-pitch interconnect chips (EFIs) in a redistribution layer-first fan-out wafer-level (FOLWLP) packaging platform, as shown in Figure 10b [54]. EFI can provide multi-layer high-density interconnects from 2 μm to 0.4 μm copper to meet the 2Gbps data transfer rate between ASIC and HBM.

In addition, fan-out packaging technology can also be considered a substrate-free vertical stacking packaging method. There is a lower cost of fan-out package implementation compared to silicon-adapter-board-based 3D packages. The package structure of the Apple A10 processor is shown in Figure 11a, which uses integrated fan-out (InFO) technology to integrate the application processor and DRAM in the same package [55,56]. InFO provides a vertical signal path from the top LPDDR package to the SoC in the InFO package via TIV (through-InFO-via). In addition, InFO features multiple layers of high-density 2/2 μm RDL line widths/spacing, allowing the integration of multiple advanced nodes of Chiplets to reduce costs and improve performance. The A10 processor is 120 times more powerful than the first-generation iPhone chip and 40% more powerful than the A9 processor in the iPhone 6s. Because RDL replaces the substrate in flip-chip packages, InFO offers the advantages of wafer-level integration, such as low latency communication between chips, high bandwidth density, and low PDN impedance. In 2019, Jhih et al. developed 3D-MiM (MUST-in-MUST) integration technology on top of InFO-based and used it to integrate 8 SoCs with 32 memory chips [57]. It was used to simulate the system integration of multiple logic chips with multiple memory chips in HPC applications, as shown in Figure 11b. The modular storage unit in this architecture integrates multiple vertically stacked memory chips via InFO. Memory modules can be embedded directly at design time, greatly reducing design complexity and shortening time-to-market. In addition, unlike 2.5D packages, InFO’s integrated memory modules allow for flexibility not only horizontally but also vertically for improved scalability.

To better leverage the advantages of Chiplet and enable system expansion with better cost and performance benefits, system on integrated chips (SoIC) technology forms a highly integrated system by implementing SoC partitioning and stacking logic chips and core circuit chips, as shown in Figure 12a [58,59]. The ultra-short interconnect distance of SoIC can scale to less than 10 μm and is capable of reaching 20 Tbps of memory bandwidth, thus enabling ultra-high-density interconnects at the chip functional block level. For example, Cheng et al. demonstrated a prototype of two CPU and memory chip stacks using SoIC technology to simulate Chiplet partitioning for high-bandwidth and low-latency applications [60]. Its findings show a 15% performance improvement and a 30% reduction in average point-to-point distance. The direct interconnection between the chips allows for the shortest distance, which not only achieves high bandwidth and low latency but also creates excellent power integrity and signals integrity. In 2021, Wei Lu et al. introduced a hybrid DNN accelerator that stacks 3D-SRAM cubes on a logic chip via SoIC technology [61]. It achieves higher than 90% PE utilization and reduces DRAM accesses by about 6× on different DNN models. The accelerator improves the overall energy efficiency by 17.4 times on VGG-16 compared to other DNN accelerators. In addition, the SoIC can be used with CoWoS and InFO to create a complete 3D stacking platform for optimal performance and cost-effectiveness, creating a more robust PIM architecture, as shown in Figure 12b.

Fan-out-package-based Chiplet is relatively low-cost compared to the silicon-adapter-board-based 2.5D/3D package. The interface between Chiplets tends to take up more area than the connection of the chips. In contrast, in a fan-out package, the I/O interfaces are rearranged in the loose area of the RDL, leaving room for integration of more Chiplets. In addition, Chiplet architectures based on fan-out packages have the smallest package size, making them the best choice for high memory bandwidth and low-power mobile applications. However, the architecture is also limited by the finer L/S and higher-level RDL requirements. Moreover, the packaging stresses caused by thermal loads may lead to RDL cracking. Since the application of InFO with Apple’s A10 processor, fan-out packaging technology has become more and more widely used in Chiplet applications. There are many fan-out packaging technologies for Chiplets integration in the industry.

### 3.4. Summary

With Moore’s Law slowing down, it is no longer desirable to add more chips to the processor to improve computing performance. In Chiplet-based PIM architecture, the interconnection distance between compute Chiplets and storage Chiplets is shortened by various advanced packaging technologies to achieve a high-speed connection with high bandwidth. Thus, a high-performance computing system is created. CEA-Leti demonstrates a 3D-based processor that stacks 16-core Chiplets on an active adapter board to create a 96-core processor [22]. It has an on-chip network that uses three different communication circuits to connect the SRAM on-chip memory on the core. The network is capable of slinging three terabytes per second per square millimeter of silicon with a latency of just 0.6 nanoseconds per millimeter. The modular IP and configurable architecture design of Chiplet allow for higher yield and cost-effective chip designs. The Chiplets are further composed into the required computing architecture through advanced 2.5D/3D packaging technology. As shown in Table 1, the NVIDIA GP100 computing platform built with 2.5D CoWoS can reach a peak bandwidth of 717 GB/S at 1.4 GHz and consume only 235 W. Agilex FPGAs using 2.5D EMIB technology not only have a high bandwidth of nearly 900 GB/s but also have reduced latency to 60 ps. Intel’s Ponte Vecchio GPU leverages 3D Co-EMIB technology to fully embrace the benefits of Chiplet. The micro-bump enables the interconnect distance to be reduced to 30 µm and the bandwidth to be increased to 2 TB/s and the architecture to be highly reusable and configurable. The Chiplet-based 2.5D and 3D PIM architectures have obvious advantages: significant improvements in integration density, data bandwidth, energy efficiency, and latency over traditional computing architectures. However, different technology configurations have different application foci. For example, 3D integration technology has greater advantages in terms of data bandwidth, but it also comes with the troubles of heat dissipation technology. In terms of cost, 2.5D integration is mostly passive adapter boards and does not require high-density TSVs, thus reducing manufacturing costs. At the same time, the heterogeneity and reconfigurability of Chiplet can be applied to different fields of PIM architecture, such as data centers, HPC, AI, etc.

## 4. Conclusions

This paper introduces the Chiplet-based PIM architecture and summarizes its implementation form and performance metrics. The Chiplet-based PIM architecture can improve the system computational performance by integrating high-performance computing chips and HBM, such as CPU/GPU/ASIC/FPGA+HBM. Combining with 2.5D, 3D, and fan-out technologies enables tighter integration of processors and memory at the package level. The ultra-short interconnect distance increases communication bandwidth and reduces latency. The Chiplet-based PIM architecture offers the following advantages: (1) it can increase throughput while reducing costs; (2) high scalability and flexibility; (3) high energy efficiency, alleviating the “storage wall”.

Driven by big data and diverse applications, the traditional computing architectures are gradually failing to meet the demand. Chiplet-based PIM architecture will become the mainstream design approach for HPC, AI, and other fields in the future. The future prospects of Chiplet-based PIM architectures are summarized as follows: (1) continue to vigorously develop new advanced packaging technologies. The application of advanced packaging technologies shows that there is still a great deal of work required in the case of unsustainable Moore’s Law. Advanced packaging technologies can better match the reconfigurability and heterogeneity of Chiplet to create a robust PIM architecture. (2) Refine Chiplet’s standard connectivity protocols. A Chiplet interconnection protocol called UCle has been developed. However, the protocol is currently defined only for 2D and 2.5D packages, and more advanced 3D technologies have not been included in the scope. Therefore, the Chiplet interconnection protocol needs to be improved as soon as possible to truly mix and match the Chiplet ecosystem.

## Figures and Tables

**Figure 1 micromachines-13-01790-f001:**
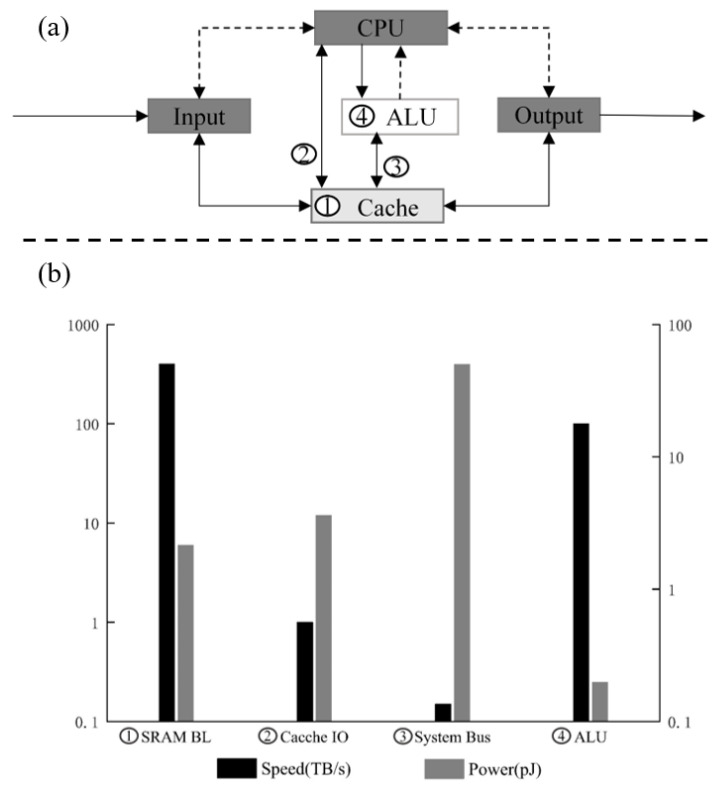
(**a**) Von Neuman architecture; (**b**) performance statistics for each module in the von Neumann architecture.

**Figure 2 micromachines-13-01790-f002:**
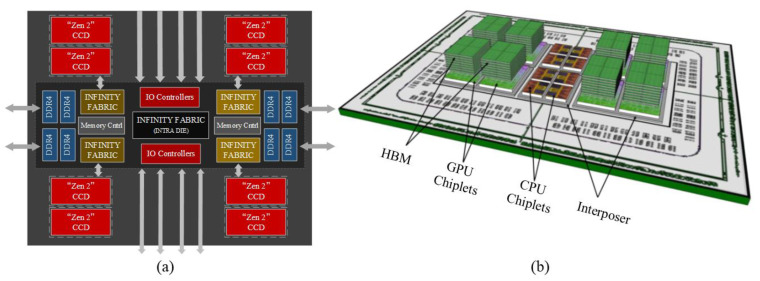
(**a**) “Rome” CPU architecture (reprinted from Ref. [26], Copyright 2021, with permission from IEEE); (**b**) Chiplet-based accelerated processing units (reprinted from Ref. [28], Copyright 2017, with permission from IEEE).

**Figure 3 micromachines-13-01790-f003:**
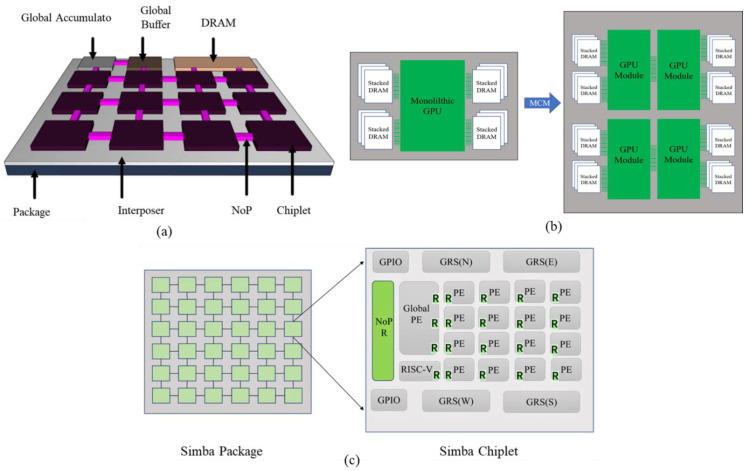
(**a**) PIM architecture based on Chiplet; (**b**) Chiplet-based design GPU (reprinted from Ref. [29], Copyright 2017, with permission from ACM); (**c**) Simba Chiplet (reprinted from Ref. [30], Copyright 2019, with permission from IEEE).

**Figure 4 micromachines-13-01790-f004:**
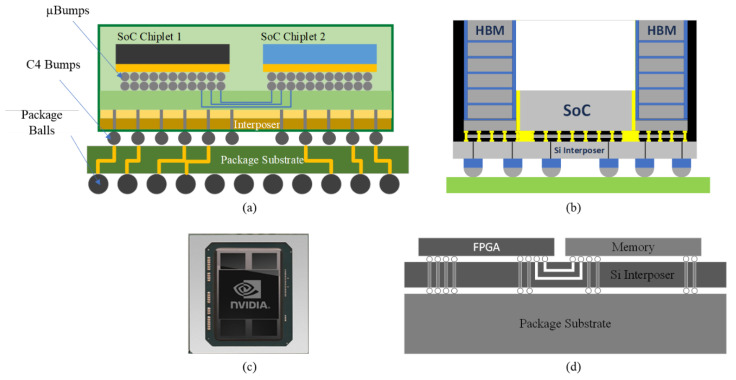
(**a**) CoWoS packaging process (reprinted from Ref. [31], Copyright 2017, with permission from IEEE); (**b**) CoWoS realizes the form of PIM (reprinted from Ref. [33], Copyright 2021, with permission from IEEE); (**c**) NVIDIA GP100 structure (reprinted from Ref. [34], Copyright 2017, with permission from IEEE); (**d**) Virtex^®^UltraScale+™ structure (reprinted from Ref. [35], Copyright 2017, with permission from IEEE).

**Figure 5 micromachines-13-01790-f005:**
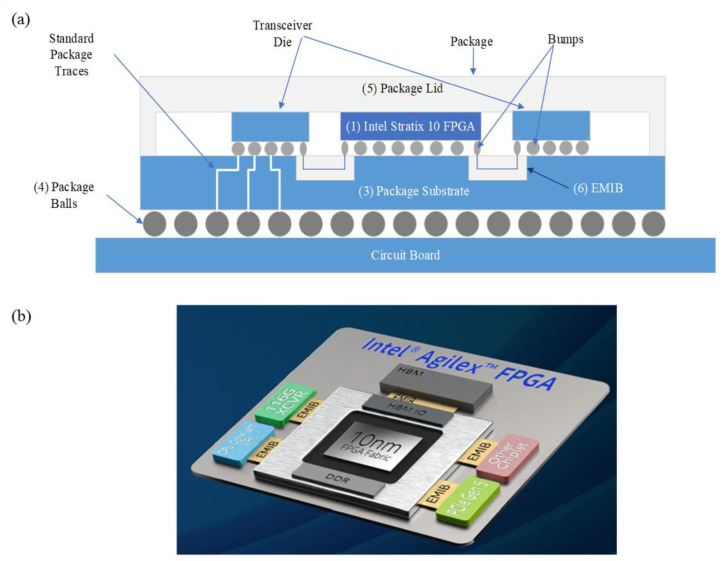
(**a**) The physical construction of the EMIB package (reprinted from Ref. [39], Copyright 2019, with permission from IEEE); (**b**) Intel Agilex FPGA structure layout (reprinted from Ref. [40], Copyright 2020, with permission from IEEE).

**Figure 6 micromachines-13-01790-f006:**
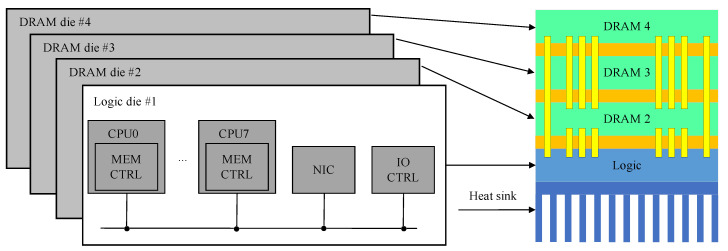
A diagram depicting the PicoServer (reprinted from Ref. [43], Copyright 2006, with permission from ACM).

**Figure 7 micromachines-13-01790-f007:**
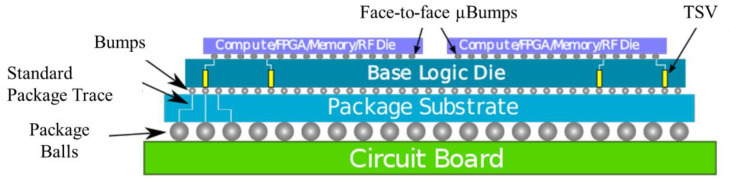
Foveros architecture (reprinted from Ref. [47], Copyright 2020, with permission from IEEE).

**Figure 8 micromachines-13-01790-f008:**
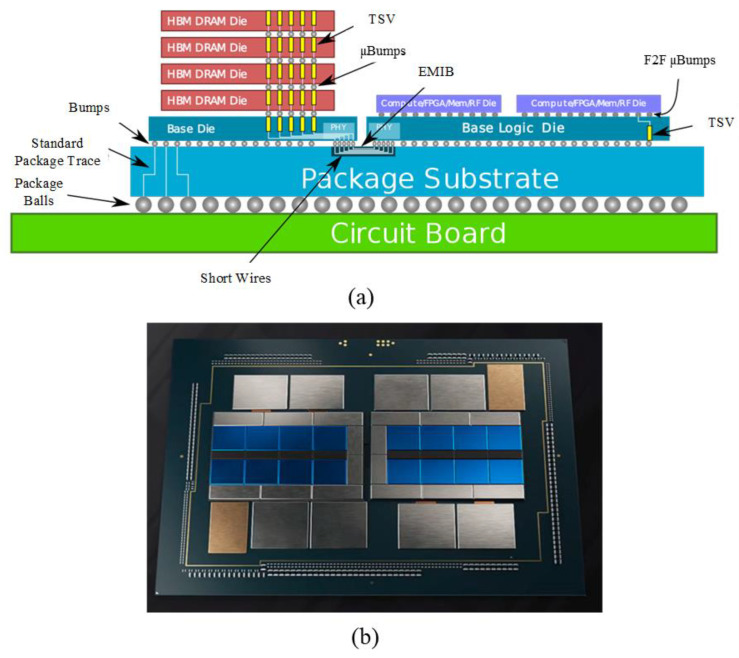
(**a**) Co-EMIB 3D Integration Technology (reprinted from Ref. [50], Copyright 2020, with permission from IEEE); (**b**) Ponte Vecchio GPU architecture (reprinted from Ref. [51], Copyright 2021, with permission from IEEE).

**Figure 9 micromachines-13-01790-f009:**
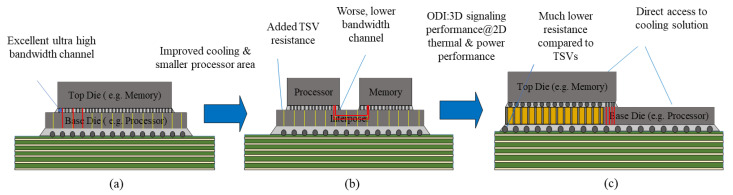
**(a)** 3D stacked processor and memory supporting large bandwidth but limited by TSV area increase & thermals; **(b)** solution of TSV & thermal problem using silicon interposer at the expense of reduced bandwidth and higher power; **(c)** ODI solution for supporting large bandwidth, good thermal performance and minimal TSV area increase. (reprinted from Ref. [52], Copyright 2019, with permission from IEEE).

**Figure 10 micromachines-13-01790-f010:**
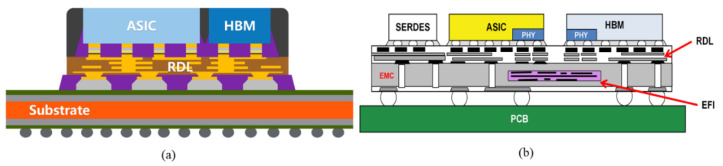
(**a**) FOCoS Technology (reprinted from Ref. [53], Copyright 2019, with permission from IEEE); (**b**) wafer-level packaging using embedded fine-pitch interconnect chips (reprinted from Ref. [54], Copyright 2018, with permission from IEEE).

**Figure 11 micromachines-13-01790-f011:**
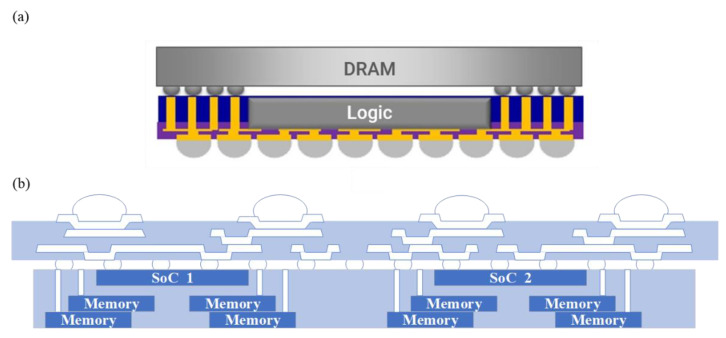
(**a**) Apple A10 chip package schematic (reprinted from Ref. [55], Copyright 2016, with permission from IEEE); (**b**) 3D-MiM (MUST-in-MUST) integration technology (reprinted from Ref. [57], Copyright 2019, with permission from IEEE).

**Figure 12 micromachines-13-01790-f012:**
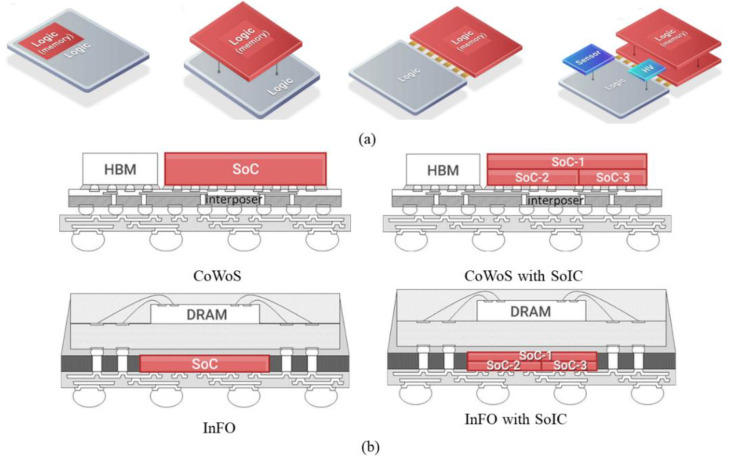
(**a**) Principle of SoIC Technology (reprinted from Ref. [58], Copyright 2019, with permission from IEEE); (**b**) compatible use of SoIC with CoWS and InFO (reprinted from Ref. [33], Copyright 2021, with permission from IEEE).

**Table 1 micromachines-13-01790-t001:** Comparison of PIM architectures based on Chiplet.

	TSMC	Intel	TSMC	CEA-Leti	Intel	Intel
Product Name	CoWoS	EMIB	InFO	INTACT	Foveros	Co-EMIB
Integrated type	2.5D	2.5D	3D	3D	3D	3D
Interposer type	Passive	Passive	-	Active	Active	Active
Interconnect pitch (µm)	40	55	-	20	36	36
PIM Application	NVIDIA GP100	Agilex FPGA	Apple A10 processor	-	Lakefield processor	Ponte Vecchio GPU
Bandwidth	717 GB/s	896 GB/s	-	527 GB/s	-	2 Tb/s
Power	235 W	-	-	~30 W	7 W	600 W
Frequency (GHz)	1.4	1.5	-	1.15	~1	1.37
Latency	-	~60 ps	-	0.6 ns/mm	-	-
Yield	High	High	High	High	High	High
Reusability	High	High	High	High	High	High
Application	HPC Edge Computing	Data Center Networking Edge Computing	Mobile IoT	HPC AI Edge Computing	Mobile PC	Date Center Machine Learning HPC

## Data Availability

Not applicable.

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
