# Peer review of "Using Chiplet Encapsulation Technology to Achieve Processing-in-Memory Functions"

_micromachines, 2022, doi:10.3390/mi13101790_

Round 1

Reviewer 1 Report

In this work, the authors seek to provide a literature review of the current status of chiplet architectures in the realm of processing-in-memory. 

The paper's strengths are that it is generally readable and addresses some new chiplet technologies that have not been covered in previous works. 

However, the paper needs a lot of work in terms of flow and sentence grammar. Specifically, there is little structure in the work that helps the reader to understand what the authors are currently talking about. Currently the only headers in the main body divide the discussing between 2.5D and 3D chiplets, with advantages and challenges discussed seemingly at random and often repeated between the two sections. Some sentences seem to be repeated near verbatim two or three times, and it is generally very hard to follow the chain of logic put forth by the work. It would be nice to have distinct sections for advantages and challenges, with subsections focused on each advantage and challenge. It is suggested to check other literature reviews for inspiration on how the paper can be structured. 

A second area of improvement that can be addressed is those of the figures. Specifically, several pictures are difficult to read in terms of their 2D-3D perspective. For example in Figure 4 b it is hard to tell if we are looking at a 2D or 3D picture, the bumps and HBM stacks imply 3D, but I don't think the SoC chip is as tall as the HBM stack. Less important, in Figure 3 there is a typo, the bumps on Figure 4 are not uniform in size, Figure 13 contains multiple typos and formatting inconsistencies and is unclear in what the different text boxes are describing, or what the flow from left to right represents.

A few other things are, first, that the acronym CoWoS is never explained. Second, as an researcher in the SRAM and also RRAM PIM field, I feel that the statement that SRAM is "generally used" in PIM chips could be debated, as non-volatile crossbar arrays are extremely popular. Also, I would personally reserve the term PIM for computation happening exclusively and physically in the memory macro, where as chiplets with computation happening in a physically different chiplet is really NMP technology.

Reviewer 2 Report

This paper discusses the problems of inmemory processing and concentrates on the implementations of Chiplets, a set of chips that comprise a larger chip, interconnected either horizontally or vertically in a specific manner, combining memory elements and processing cores in a heterogeneous architecture. The paper presents numerous such implementations, mostly from industry, from the last 5 years.

The paper is well presented and documented; there is no paper outline at the end of Section 1 though.

It has minor grammatical and reference connection errors in the main text that need to be fixed.

Its main weakness is that it fails to provide a full coverage of research directions on PIM technology, spending about only half a page for such review (within Section 2), although the first part of the paper was supposed to present such a review (as suggested by Abstract and title of Section 2).

It is suggested that the specific part of the paper is rewritten to include a more in-depth review on PIM technology research.

Round 2

Reviewer 1 Report

Hello and thank-you for the revised version of this work. I have looked over the changed sections and appreciate the large amount of clarifying text added, as it does clarify some of my questions.

Unfortunately, the structure of this paper still is not clear as a literature review, as breaking chiplets into only 2 categories of 2.5D and 3D is not sufficient to help a reader understand the current state, advantages, disadvantages, and future opportunities and challenges of chiplet technologies.

The added sections do provide some clarification regarding the content, especially with clarifying CoWoS technology, however, the new sections also contain numerous typos and sentence fragments. Further, somehow now a typo has snuck into Section 3.2's header, and there is a missing reference on page 6. Therefore, the work still needs a major overhaul in structure and grammar before it can be publicly shared. 
